# Clinical Presentation, MRI Characteristics, and Outcome of Conservative or Surgical Management of Spinal Epidural Empyema in 30 Dogs

**DOI:** 10.3390/ani12243573

**Published:** 2022-12-17

**Authors:** Carlos Blanco, Meritxell Moral, Juan José Minguez, Valentina Lorenzo

**Affiliations:** 1Neurología Veterinaria, Calle Diseño n 26, 28906 Getafe, Spain; 2Scarsdale Vets-Pride Veterinary Centre, Riverside Road, Derby DE24 8HX, UK

**Keywords:** canine, nervous system, infection, spine, surgery

## Abstract

**Simple Summary:**

Spinal epidural empyema (SEE) represents a neurological emergency in veterinary medicine, but information on this condition is still scarce. The aim of this retrospective study was to describe the clinical presentation, magnetic resonance imaging (MRI) features, and outcome after conservative or surgical treatment of 30 dogs with SEE. Although dogs affected by this condition require prompt intervention, clinical signs may be non-specific, and diagnosis may be delayed. MRI features can help in timely diagnosis and follow-up management. Favourable outcomes can be obtained in surgically and conservatively treated dogs, even in cases with severe signs and epidural compression.

**Abstract:**

Spinal epidural empyema (SEE) represents a neurological emergency in veterinary medicine, but information on this condition is limited to date. This retrospective case series study describes the clinical and magnetic resonance imaging (MRI) features, and the outcome of conservative or surgical management of SEE in 30 dogs diagnosed with SEE from September 2015 to March 2020 at one referral neurology centre. The most frequent clinical sign was pain 28/30 (93%), and 22/30 (73%) showed neurological signs with ambulatory paraparesis/tetraparesis 15/30 (50%), monoparesis 1/30 (3.3%), non-ambulatory paraparesis 3/30 (10%), or paraplegia 3/30 (10%). MRI was valuable for the diagnosis and in the follow-up. In this group of dogs, 24/30 (80%) were conservatively treated and 6/30 (20%) were surgically treated. The outcome was considered favourable in all dogs: 20/30 (66.6%) achieved full recovery (3 surgically treated and 17 medically treated) and 10/30 (33.3%) dogs had an improvement in the neurological signs with residual ambulatory paresis (3 surgically treated and 7 medically treated). Surgical treatment showed better short-term (7 days) outcomes than medical treatment in non-ambulatory paraparetic or paraplegic dogs (33%). Nevertheless, this study suggests that a good recovery may be achieved with conservative treatment even for non-ambulatory or paraplegic dogs. Further prospective studies, with a standardised protocol of diagnostic tests and a homogeneous distribution of conservatively and surgically treated dogs, are needed to establish treatment guidelines.

## 1. Introduction

Spinal epidural empyema (SEE) refers to a non-encapsulated accumulation of purulent material in the epidural space [1,2], representing an infectious process coming from several routes that may include hematogenous spread from distant infections, spread from adjacent tissues, direct trauma, foreign bodies, and invasive spinal procedures such as surgery [1,3,4,5]. The pathophysiologic mechanism of the neuronal damage associated with spinal epidural empyema is controversial [6]. A primary mechanical compression with secondary vascular compromise appears to be the most likely pathologic mechanism [1]. In human medicine incidence has increased in recent years [4,7], but the incidence in veterinary medicine remains unknown [3,8]. Initial clinical signs include spinal pain, fever, and progressive neurological dysfunction [1,8]. SEE has been categorized as a neurological emergency that requires prompt surgical or medical intervention to prevent permanent neurological sequelae [9,10], but as clinical signs in both human and veterinary medicine may be non-specific, diagnosis may be delayed [5,8,11]. In this situation, magnetic resonance imaging (MRI) is highly valuable for prompt diagnosis as it is the gold standard technique for assessing the spinal cord and adjacent structures [1,12,13,14]. While early diagnosis and management are important to optimize the outcome, the approach to managing this condition remains undefined [3,4,5,7,8,11]. A recent study showed no significant long-term difference between medical or surgical treatment [12], suggesting that conservative treatment may be appropriate for ambulatory dogs on admission.

This retrospective case series aimed to describe the clinical presentation, the MRI characteristics, and outcomes after medical or surgical management in 30 dogs diagnosed with SEE, between September 2015 and March 2020 at one referral neurology centre. Our premise was that conservative treatment might have a favourable outcome comparable to surgical treatment, even for dogs with severe neurological deficits. 

## 2. Methods

A retrospective descriptive case series included dogs diagnosed with SEE examined from September 2015 to March 2020 at one referral neurology centre. Medical records and MRI studies with the corresponding reports were searched. The information retrieved included signalment, history, pre-referral of complementary tests (blood tests, x-rays, ultrasound) and previous treatments, general physical examination, neurological examination findings, imaging findings, cerebrospinal fluid (CSF) cytology, results from CSF and urine culture (if performed), medical treatment, surgical procedure if performed, follow-up including MRI results if performed, and outcome. To be included in the study, dogs had to fulfil the following inclusion criteria: an MRI scan showing a space-occupying accumulation of epidural material suggestive of empyema, and confirmation of the diagnosis either by CSF analysis, direct visualization of purulent material during surgery, or response to treatment with antibiotics. 

The neurological status at the time of presentation and during treatment, at the time of discharge, and at re-examinations every three weeks was recorded. All neurological examinations and follow-ups were performed by a residency-trained or board-certified neurologist. The neurological status was scored using a six-point grading scale (modified Frankel Scale [15]): grade 0 (neurologically normal), grade 1 (painful, no neurological deficits), grade 2 (ambulatory paraparetic/ataxic), grade 3 (non-ambulatory paraparetic), grade 4 (paraplegia intact deep pain perception, and presence of superficial pain and involuntary urination), or grade 5 (paraplegia without nociception). Response to a treatment referred to functional improvement according to the modified Frankel Scale [15]. Follow-up had to be for a minimum of 3 months. The outcome of both surgical and medical treatments in the short term (less than or equal to 7 days) was considered successful if the dog improved at least one neurological grade. Long-term outcome (from 3 to 6 months) was considered successful if the dog was independently ambulatory, urinary and fecally continent, and not in pain. 

## 3. MRI and CSF Acquisition and Evaluation

MRI imaging studies of the vertebral column were acquired with a 1.5 Tesla unit (Gyroscan Intera, Philips, Eindhoven, The Netherlands). All studies were performed with the dogs in dorsal recumbency under inhalation anaesthesia. Images included, at least, transverse and sagittal T2-weighted (T2W) sequences and T1-weighted (T1W) sequences before and after contrast administration (gadoteric acid 0.1 mmol/kg IV, Dotarem, Guerbet). MRI diagnostic features were based on the characteristics of SEE in previous descriptions [4,16]. MRI changes consistent with SEE included the presence of space-occupying accumulation of epidural material (either focal, multifocal, or diffuse) characterised by a hyperintense or mixed signal on T2W sequences when compared to the spinal cord, a hypointense signal on T1W sequences, and the presence of postcontrast enhancement on T1W sequences with variable patterns (diffuse, heterogeneous or ring-like). The lesion’s extension over the vertebral bodies’ length and the degree of occupation of the vertebral canal diameter (mild, <25%; moderate, 25–50%; severe, >50%) were also considered. 

Follow-up MRI studies were performed when possible. Each MRI study on presentation and its subsequent follow-up were reviewed by two authors, a residency-trained neurologist, and a board-certified neurologist.

CSF tap was performed in selected cases, considering the possibility of infection spreading. In cases in which CSF analysis was accomplished, CSF was collected immediately after MRI by cerebello-medullary cistern or lumbar puncture. CSF was evaluated for cellular content and morphology, and microproteins.

The considered reference values for total nucleated cell count (TNCC) were ≤5 cell/µL, and for total protein value < 25 mg/dL or <45 mg/dL cisternal or lumbar, respectively. CSF analysis was considered compatible with SEE if the results met the previously reported criteria [8,17] of neutrophilic pleocytosis, and subsequent culture was carried out, if possible, with 0.5–0.8 mL of CSF or with 2–5 mL of urine.

## 4. Statistical Analysis

First, we used a Shapiro–Wilk test to examine the normality of residuals. We then conducted an a priori power analysis to determine the smallest sample size suitable to reject the null hypothesis, given our goal to obtain a 0.80 power at the 0.05 alpha to detect at least a large effect size d = 0.80 while considering an allocation ratio of 24 to 6. 

## 5. Results

Thirty dogs were included in the study. The median age was 6.1 years (range of 8 months-11 years). Sixteen dogs were male (9 neutered) and 14 were female (10 neutered). The average weight was 24.4 kg (range 3.6–50 kg). Breeds comprised American Staffordshire Terrier (n = 5), Labrador Retriever (n = 3), Doberman (n = 2), Dalmatian (n = 1), Yorkshire Terrier (n = 2), German Shepard (n = 2), Golden Retriever (n = 2), French Bulldog (n = 1), West Highland white terrier (n = 1), English Bulldog (n = 1), Maltese (n = 1), crossbreed (n = 5), Greyhound (n = 2), English Pointer (n = 1) and Dachshund (n = 1). An acute (sudden and severe) and progressive onset was reported for all dogs. At presentation, a general physical examination revealed abnormalities in 28/30 dogs, including pain (28/30), pyrexia (3/30), chronic dermatitis (1/30), and skin wounds (1/30). Neurological examination showed abnormalities in 22/30 dogs, which included ambulatory paraparesis (13/22), non-ambulatory paraparesis (3/22), paraplegia with preserved deep pain sensation (3/22), ambulatory monoparesis (1/22), ambulatory tetraparesis (1/22) and nerve root sign (1/30).

Eight of the thirty dogs had been diagnosed with eight different pathologies between eight weeks and ten days before presentation, which included: radius fissure, surgical removal of a subcutaneous nodule, prostatitis, Ehrlichiosis, panniculitis, pneumonia, Leishmaniosis, and skin wounds. One of them was on immunosuppressive treatment for panniculitis. Twenty-seven of the thirty dogs had been receiving therapy before referral for the signs related to SSE, which included: glucocorticoids (6/27), non-steroidal anti-inflammatory drugs (NSAID) (4/27), glucocorticoid plus analgesics, and muscle relaxant (3/27), antimicrobials (one dog with cephalexin −22 mg/kg BID (*bis in die*, twice a day))- and two dogs with amoxicilin/clavulanic −22 mg/kg BID-) plus NSAID and analgesics (3/27), NSAID and analgesics (6/27), antimicrobials (one dog with cephalexin 22 mg/kg BID and one dog with metronidazole 10 mg/kg TID (*ter in die*, three times a day)) and NSAID (2/27), antimicrobials (one dog with metronidazole 10 mg/kg TID and one dog with amoxicilin/clavulanic 22 mg/kg BID) plus glucocorticoids and analgesics (2/27), and antimicrobials (amoxicilin/clavulanic 22 mg/kg BID) (1/27). All dogs showed mild and transitory improvement. The time from onset of clinical signs to remission ranged from 2 days to 4 months. 

In the tests carried out before referral, a complete blood cell count (n = 27) showed neutrophilia (5/27), anaemia (1/27), and thrombocytopenia (1/27). Biochemistry (n = 27) showed hyperglobulinemia (3/27), hypoalbuminemia (1/27), and elevation of ALT (2/27). C-reactive protein tested in one dog was increased (90.7 mg/dL, reference range 0–10 mg/dL). Urinalysis (2/27) was performed with negative results for bacterial culture. Echocardiography (n = 1) showed mitral regurgitation B1 (according to the stages of myxomatous mitral valve diseases). Abdominal ultrasound (n = 8) showed prostate microabscesses (1/8) and adrenal gland hyperplasia/neoplasia (1/8). Thoracic radiography (n = 7) showed a bronchial pattern in one of seven dogs. Spinal radiography (n = 9) findings included changes compatible with spondyloarthritis (3/9), intervertebral collapse at L3-L4 (1/9), and osteolysis in the body of T3 (1/9). Abdominal radiography (n = 4) demonstrated a foreign body in the stomach of one of four dogs.

### 5.1. MRI Features at the Presentation

All dogs had an MRI of the spine within 24 h of presentation (Figure 1A–C, Figure 2A and Figure 3A–D). The images revealed the presence of a space-occupying accumulation of epidural material suggestive of empyema. The epidural material was hyperintense or had a mixed (hyperintense and isointense) signal on T2W sequences compared to the spinal cord and had an iso to hypointense signal on T1W sequences. After contrast administration, enhancement was observed in all cases with variable patterns (diffuse, heterogeneous, or ring-like). 

The images revealed the presence of a space-occupying accumulation of epidural material suggestive of empyema localised in the spinal segments C1–C5 (6/30), C6–T2 (2/30), T3–L3 (11/30), or L4–S3 (11/30). In 7/30 dogs the epidural material was extensive, involving from two to up to nine consecutive vertebral bodies. In addition, 15/30 dogs had signs of discospondylitis at adjacent levels, and in 13/30 dogs there was involvement of adjacent soft tissues and vertebral bone. Only in 2/30 dogs was empyema the only finding. 

### 5.2. Clinicopathological Analysis

CSF analysis performed in 19 dogs showed pleocytosis in 11/19 (range 7–9300 cel/µL), with a neutrophil count range between 50% and 85%. Total protein was increased in 17/19 (range 32–400 mg/dL). CSF was normal in two dogs. CSF culture (n = 4) yielded growth of *Streptococcus Canis* (1/4) and urine culture (n = 13) of *Escherichia Coli* (1/13).

In 6/30 dogs, surgery was performed to alleviate spinal cord compression and to obtain samples for culture. In this group, as the material obtained was a purulent liquid, histopathological studies were not considered, but the samples were submitted for microbial culture. All the samples yielded bacterial growth and included *Pseudomona aeruginosa* (2/6), *Aerococcus* spp. (1/6), *Serratia marcescens* (1/6), *Burkholderia cepacia* (1/6), and *Streptococcus beta-haemolytic* (1/6). Reported treatment prior to culture included doxycycline for 1 month (1/6), antimicrobial and cyclosporine for 1 month (1/6), and glucocorticoids at immunosuppressive dose of 1 mg/kg for 10 days (1/6). 

### 5.3. Treatment, Follow-Up and Outcome

Following the presumptive diagnosis of SEE, 24/30 dogs were medically managed and 6/30 received surgical treatment. Surgical treatment was offered to non-ambulatory or paraplegic dogs or those with extensive (extending to two or more vertebral bodies) epidural lesions. The final decision for treatment was achieved by consensus with the owners after they were informed of the treatment options, namely surgical and medical treatment, or medical treatment alone.

In the conservatively treated group, the most severe cases included one dog with non-ambulatory paraparesis and two dogs that had paraplegia with preserved deep pain sensation for 24–48 h. Regarding the imaging, 7/24 had extensive epidural occupation with involvement of two intervertebral levels up to nine consecutive vertebral bodies, or had spinal cord compression of 25–50%. If the dogs had a previous antimicrobial treatment on admission, the same regimen was maintained and combined with other antimicrobials to broaden the spectrum. Fourteen dogs were hospitalised and received a combination of trimethoprim sulfonamide (15 mg/kg/BID, IV) and metronidazole (10 mg/kg/TID, IV) (10/14), or amoxicillin/clavulanic acid (22 mg/kg BID), enrofloxacin (5–10 mg/kg SID) and metronidazole (10 mg/kg TID) (4/14). All the dogs were discharged between three and seven days. Ten of the medically treated group dogs were not hospitalised due to financial concerns. They received oral treatment with a combination of enrofloxacin (5–10 mg/kg SID), trimethoprim sulfonamide (15 mg/kg/BID) and metronidazole 10 mg/kg/TID (6/10); amoxicilin/clavulanic (22 mg/kg BID), enrofloxacin (5–10 mg/kg SID) and metronidazole (10 mg/kg TID) (2/10); and cephalexin (22 mg/kg BID) and metronidazole (10 mg/kg TID) (2/10). Prednisolone (0.5 mg/kg SID or BID) was administered in eight dogs (range 3–10 days) as they were already on glucocorticoids (dose range 0.5 to 1 mg/kg SID or BID for 7–15 days). The two dogs that presented positive cultures either in CSF (*Streptococcus Canis)* or urine (*Escherichia Coli*) were treated according to the sensitivity results. Neurological improvement was evident between 48 h and 14 days in this medically treated group. Two dogs required extra analgesia treatment due to radicular pain. Analgesics (methadone 0.2 mg/kg/4 h for the first 24–36 h, then paracetamol 10 mg/kg BID) were administered until dogs were considered to be non-painful on recheck examinations. In the follow-up, antimicrobials were progressively withdrawn every 4 weeks, and as long as there were no signs. Metronidazole was discontinued first, then enrofloxacin, and finally trimethoprim-sulfonamide or amoxicillin clavulanic acid.

In the conservatively managed group of dogs, 7/24 of them met the criteria for surgical treatment as they had either an extensive epidural occupation with involvement of two intervertebral levels up to nine consecutive vertebral bodies (5/7), or had an occupation of the spinal canal of 25–50% (2/7), or were non-ambulatory paraparetic (1/7) or paraplegic for 24–48 h with preserved deep pain sensation (2/7). However, due to financial constraints, they could not be treated surgically. 

In the group of 6/30 dogs surgically treated, the surgical technique was chosen to take into account the best visualisation of the lesion and decompression, with hemilaminectomy (3/6), pediculectomy (1/6) or dorsal laminectomy (2/6). One dog temporarily worsened, changing from ambulatory paraparetic to non-ambulatory paraparetic over three days, and then improved; the rest showed progressive improvement. Awaiting the result of the cultures, all cases received antimicrobial treatment with trimethoprim sulfonamide (15 mg/kg/BID IV) for 3–5 days. After the culture results from the surgical swabs, antimicrobials were modified in 4/6 dogs. The median antimicrobial course length in all dogs was 16 weeks (range 12–20 weeks).

Follow-up MRI studies were performed (n = 13) from 2 to 5 months after treatment initiation, in 8/13 dogs conservatively treated (Figure 1D–F) and in 5/13 of the dogs surgically treated (Figure 2B and Figure 3E–G). Radiological improvement was noticed in all dogs, considering there were no signs of epidural material or soft tissue enhancement. Bone changes (osteoproliferation and mild contrast enhancement) were still present.

Regarding the outcome, 20/30 dogs achieved full recovery (three surgically treated and seventeen medically treated), and 10/30 dogs improved but maintained residual ambulatory paresis during the time of follow-up (three surgically treated and seven medically treated). Within the group of 10 dogs with residual signs, 3/10 presented with paraplegia on admission, and 2/10 were medically treated. One medically treated dog with initial tetraparesis that improved to ambulatory paraparesis was euthanised because of a hepatic carcinoma diagnosed 3 months later, and was considered in the group of residual paraparesis. 

The period of clinical improvement for both the conservatively and the surgically treated groups is shown in Table 1.

### 5.4. Statistical Analysis

Having determined the non-normality of residuals using the Shapiro–Wilk test (*p* = 0.003), an a priori power analysis for a Mann–Whitney U test (two-tailed) yielded a minimum sample size of at least 17 dogs in the surgery condition and 67 in the medical condition, thus totalling 84 dogs, whereas the current sample size in this study was, respectively, 6 and 24 dogs. Therefore, the study is presented as a descriptive case series.

## 6. Discussion

Spinal epidural empyema (SEE) is a life-threatening infectious condition. Still, to date in veterinary medicine, the incidence and mortality are not well established [3,8]. The number of studies on SEE [3,4,8,11,12,16] and intracranial empyema [18,19,20,21] published in the last 15 years remains relatively low. In human medicine, the incidence of hospital admission for this condition has tripled [5], and the mortality rates are estimated at 5–16% worldwide, with less than 50% of surviving patients achieving full recovery [1,22,23]. 

Given the mismatch between the minimum sample size required for a significative statistical analysis and that in this study, we were unable to conduct inferential tests and thus decided to present this research as a descriptive case series. The mean age at presentation of signs was six years, and SEE was more frequent in entire males (43%) compared to entire females (28%), coinciding with a recent study [12]. Regarding risk factors, no predisposing factors to this pathology have been described in veterinary medicine [4,8,11,12]. However, in our case series, 7/30 dogs (23.3%) had previous pathologies, including an infectious process, surgery, or immunosuppressive treatment. In human medicine, these, together with diabetes, are considered risk factors [5,22]. Therefore, previous pathologies seem to be an important factor to consider in dogs with SEE. In our study, for the case with Streptococcus beta-haemolytic infection, an hematogenous spread from the genitourinary system was suspected, as the dog had undergone orchiectomy for prostatitis 10 days before the presentation of signs; for the rest, the primary cause remained undetermined.

In our study, the most common clinical signs at admission were spinal pain (93.3%) and pyrexia (10%), in accordance with other publications [4,11,12]. In human medicine, most of the patients also present back pain (71%) and pyrexia (66%) as the initial symptoms [1]. Apart from these signs, a progressive neurologic dysfunction has been reported in both humans and dogs with SEE [3,4,5,12]. In human medicine, this triad is only present in 10–30% of initial presentations and correlates to advanced stages, which may lead to a significant diagnostic delay in about 75% of patients [5]. Similarly, in our group of dogs, only 10% of them presented the triad, therefore, in an attempt to avoid diagnostic delays, this pathology must be considered despite not having the reported triad of signs. 

Regarding laboratory findings, on admission, 18.5% of the dogs in our study had neutrophilia, different from previous studies that report higher numbers (59–85%) [4,8,11,12]. In humans, the prevalence of leukocytosis with neutrophilia is 60–78% [24,25]. A possible explanation for our lower percentage could be the administration of antimicrobials before referral. This same pre-remission administration of antimicrobials may have affected the results of our urine 1/13 (7.6%) and CSF 1/4 (25%) cultures, although it must be considered that CSF is reported to be often culture-negative [8,12].

All six samples collected during surgery yielded bacterial growth, with 2/6 cases positive for *Pseudomona aeruginosa* and each of the rest for *Aerococcus* spp., *Serratia marcescens*, *Burkholderia cepacia*, and *Streptococcus beta-haemolytic*. The only pathogen in common with previous studies is Pseudomona aeruginosa [12]. In human patients, Gram-positive Staphylococcus aureus is the most common agent [1,5,26]. Our case series shows increased identification of causative agents in surgical specimens as described in a recent SEE study [12] and human medicine [26,27]. The results obtained in the present case series do not allow a conclusion on the most common causative agent due to the low number of positive cultures and the variability of the results.

Blood C-reactive protein (CRP) has been used as a non-specific marker of inflammation. In one study it was increased in 50% of dogs with discospondylitis, at the same range of steroid-responsive meningitis-arteritis and immune-mediated polyarthritis [28]. In human medicine, the measurement of CRP levels has been found to shorten the diagnostic delay of patients with spinal infections, including SEE [5]. Although CRP was only measured in one dog in this retrospective study, based on recent studies [28], it should be part of the diagnostic protocol in cases suggestive of SEE.

SEE diagnosis should always be supported by imaging, with MRI remaining the most reliable method [1,5,12,23]. MRI is the gold standard for the diagnosis of spinal infections, due to its high sensitivity (96%), high specificity (94%), and capability to provide detailed data on paraspinal tissues and the epidural space [1,12,13,14]. SEE was more commonly found in the thoracic and lumbar spinal cord segments (73.2%), similar to a recent study [12]. This trend could be related to the high incidence of discospondylitis in the lumbar region [29] and the proximity to the urogenital tract, which may act as a source of infectious pathogens. In most of our dogs (93.4%), soft tissue and bone involvement were observed; in the rest SEE was the only finding. Discospondylitis or osteomyelitis have been frequently described in association with SEE [4,8,30]. In our case series, discospondylitis was found at adjacent levels in 50% of the dogs, in correspondence with the findings reported in previous reports [4,8,12]. In humans, a correlation has been found between the nature of epidural inflammation and the chronicity of the symptoms, with a changing pattern that can be reflected in MRI. An acute clinical presentation with symptoms for fewer than 16 days was more likely to be associated with a largely purulent epidural collection (abscess), and a chronic presentation was more likely to be associated with vascularised granulation tissue (phlegmon). On post-contrast MRI sequences, uniform enhancement is seen as there is no purulent component. In contrast, an abscess is a rim-enhancing hypointense collection [31]. The low-intensity signal on the T1-weighted images correlates with the liquid purulent component, which represents the necrotic centre of the abscess. It is a relatively inaccessible extravascular space with a low accumulation of contrast material. The surrounding rim of granulation tissue is well-perfused and enhanced following gadolinium injection [32]. This abscess pattern was also present in 7/30 dogs in the cases included in the present study which had clinical signs for more than 16 days. Therefore, this radiological pattern can be helpful when interpreting MRI studies.

Regarding treatment, the choice of medical versus surgical management of SEE remains debatable in veterinary and human medicine [1,3,4,5,8,12]. In the latter, the decision to establish medical management is based on one or more of the following criteria [11,33,34]: absence of substantial neurological deficits, non-compressive spinal injuries, extensive injuries for which multiple decompressive surgical procedures would create a significant risk of spinal instability, poor anaesthetic or surgical candidates, and tetra/paraplegia for >48 to 72 h.

In human patients, the rate of failure with conservative management ranges from 30 to 40% [35,36], but this is not well established in veterinary medicine. In this case series, of the 24/30 dogs that were treated medically, 7/24 had an extensive epidural mass, and two of them presented with paraplegia for 24–48 h, but all had a good outcome. Further studies are needed to establish a guideline for the medical treatment of these cases; however, our results suggest that medical treatment may also be considered in cases with extensive lesions or severe neurological signs. The medical treatment used in this case series for dogs in which the causative agent was not identified was a combination of antimicrobials based on our knowledge of the situation and our observations. 

Surgical treatment in this case series was indicated for dogs with ≥25–50% spinal cord compression. In human medicine, surgical treatment is established based on the degree of compression of >50% on cross-sectional MRI images [37], which correlates with severe motor neurological deficits. This approach shows favourable results in patients with deficits of less than 36 h duration [38].

Long-term results offered no significant differences between medical and surgical treatment. However, in the short term, the data suggest that surgical cases show an earlier response (up to 7 days) than those treated medically in the same time frame. This result is consistent with a recently published study on 41 dogs [12]. However, the risks of general anaesthesia and surgery must be considered, as well as the possible favourable outcome with non-surgical treatment. A delay in diagnosis may allow a progression of clinical signs and may be associated with devastating consequences despite adequate treatment [3,5,8]. Nevertheless, although early diagnosis in SEE is important to avoid sequelae, our study suggests that some of the dogs that present with non-ambulatory paraparesis and chronic signs may recover fully either with medical or surgical treatment.

In the present case series, follow-up MRI studies were performed on 13 dogs between 2 and 5 months after the initiation of treatment. The imaging showed radiological improvement in all cases, with the absence of previous signs of empyema and osteolysis, and a decrease in paraspinal soft tissue swelling. In the human literature, a correlation between the MRI findings at follow-up and the clinical status has been studied [39], and although no single MRI parameter was associated with the clinical status of patients, it was concluded that soft tissue findings, not bone findings, should be the focus of clinicians interpreting follow-up MRI results. A decrease in soft tissue findings (epidural enhancement, paraspinal inflammation, and disk space enhancement) was considered to be a good indicator of improvement, and the patients whose MRI was equivocal or worse consequently had their antibiotic course prolonged or more frequent invasive intervention than the patients with improvement on follow-up MR imaging [39]. A follow-up MRI is advised in cases with SEE as it provides valid information on the outcome of these patients.

## 7. Conclusions

SEE is a neurological emergency that requires early diagnosis and appropriate treatment, as it is potentially fatal. Sadly, a delay in diagnosis is not uncommon due to the unspecific clinical signs.

MRI represents a valuable tool for timely diagnoses and follow-up management. 

Surgical treatment, considering the limited number of cases, tends to lead to more rapid clinical improvement than conservative treatment in patients with more severe clinical signs. The difference between conservative and surgical treatment in the long term does not appear to be significantly different, even for patients with chronic signs and severe neurological deficits. 

The information provided here and added to the previous reports may contribute to better management of this challenging pathology in terms of risk factors, diagnosis, treatment, and outcome. Further studies are needed with larger and more homogeneous numbers of surgically and conservatively treated dogs, and with full complementary tests and haematological cultures, in order to identify the most common pathogens and to elaborate on treatment guidelines. 

## 8. Limitations

This is a retrospective study analysing cases seen within a large time frame. Ideally, CRP would have been measured in all cases, and haematological cultures obtained to identify pathogens and avoid antimicrobials combinations that may cause bacterial resistance. Additionally, the number of surgical cases is, compared to the conservatively treated cases, very small. However, we believe that this analysis of seen cases helps in understanding and treating these cases.

## Figures and Tables

**Figure 1 animals-12-03573-f001:**
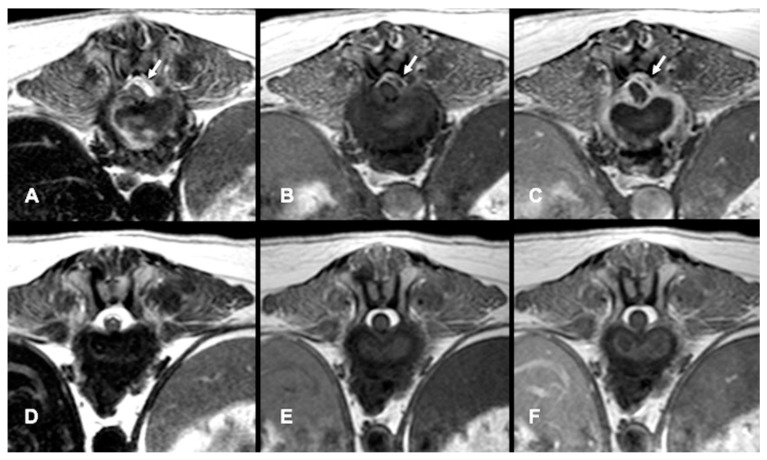
MRI transverse planes at the level of T13-L1 intervertebral disc at time of presentation (**A**–**C**) and follow-up at 4 months (**D**–**F**) in a dog diagnosed with SEE conservatively treated. A left dorso-lateral epidural lesion hyperintense on T2-W (arrow, **A**) and hypointense on T1-W (arrow, **B**) is displacing the spinal cord. On the T1-postcontrast image there is a ring enhancement of the epidural lesion (arrow **C**) and irregular soft tissue enhancement around the intervertebral disc, which shows irregular shape, and heterogeneous hyperintense signal on the T2-W image (**A**). On the follow-up images on T2-W (**D**), T1-W (**E**) and T1-W postcontrast (**F**) there are no signs of the previous epidural lesion nor the contrast enhancement.

**Figure 2 animals-12-03573-f002:**
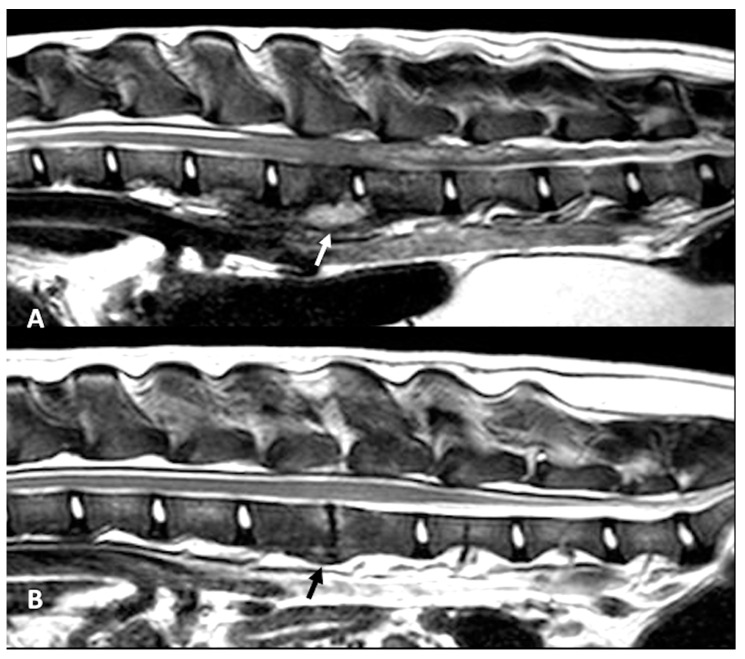
MRI T2-W sagittal plane at time of presentation (**A**) and follow-up at 4 months (**B**) in a dog with SEE surgically treated. On the intervertebral disk L3–L4 there is an infiltrative lesion also affecting the ventral paravertebral muscles and vertebral body of L3 (white arrow). The follow-up image (**B**) demonstrates the resolution of the paravertebral lesion and residual changes with new bone formation, loss of intervertebral space and mild vertebral marrow edema (black arrow).

**Figure 3 animals-12-03573-f003:**
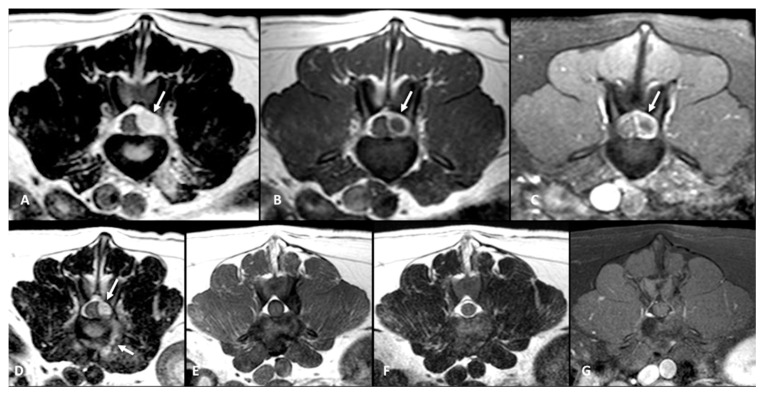
MRI transverse planes T2-W (**A**), postcontrast T1-W (**B**), and postcontrast fat-saturated T1-W (**C**) of the same dog as Figure 2, showing a left dorso-lateral epidural space-occupying lesion (arrows) in the L3–L4 intervertebral level with severe spinal cord compression. Transverse T2-W image of the same dog at L4–L5 (**D**) also demonstrates an infiltrative lesion that extends through the left intervertebral foramen (lower arrow) in connection with the epidural lesion (upper arrow). Follow-up transverse T1-W (**E**), T2-W (**F**) and postcontrast fat-saturated T1-W (**G**) images at 4 months after surgery depict the resolution of the infiltrative foraminal and post-foraminal lesions and of the epidural.

**Table 1 animals-12-03573-t001:** Short-term and long-term outcome of 30 dogs conservatively and surgically treated for SEE.

Time of Clinical Improvement	Medically Treated	Surgically Treated
7 days	13/24 dogs (54.1%)	6/6 dogs (100%)
1 week–3 months	11/24 dogs (45.9%)	

## Data Availability

The datasets used and analysed during the current study are available from the corresponding author upon reasonable request.

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
