# Peer review of "Clinical Presentation, MRI Characteristics, and Outcome of Conservative or Surgical Management of Spinal Epidural Empyema in 30 Dogs"

_animals, 2022, doi:10.3390/ani12243573_

Round 1

Reviewer 1 Report

Summary

The aim of the retrospective study is to describe the clinical presentation and outcome of dogs with spinal epidural empyema (SEE) using magnetic resonance imaging (MRI) as the diagnostic tool in a referral centre during a 5-year period. Thirty dogs met the inclusion criteria and were included. Clinical signs were non-specific and included pain, pyrexia and neurological signs. Dogs were divided into two treatment groups, 24 received conservative management and 6 were treated surgically. There were no significant differences in short-term and long-term outcomes between conservatively and surgically, although at short-term, surgically treated dogs responded faster. Also, severely affected dogs treated conservatively showed improvement suggesting that conservative management could be a good option in cases of financial constraint or associated risk factors.

Comments

Thank you so much for conducting this retrospective study. Only a few reports of this condition are available, and more information and studies are needed. It is valuable that you have included MRI in the follow-up and the way the data has been presented in the tables is very useful. Infection can result from either the hematogenous spread of bacteria or by direct local extension, could you determine the source and route of infection in the cases? You mentioned that eight of the 30 dogs had been diagnosed with eight different pathologies between eight weeks and ten days before presentation. Any correlation with the SEE?

Any correlation between the follow-up results of the MRI and the improvement of clinical signs? Any prognostic utility?

Specific comments

Line 76: did conservative management include any debridement of any paraspinal abscess?

Line 85: Were urinary and faecal continence assessed? If so, how many dogs were showing signs of incontinence?

Line 116, 231: Spelling mistake, fot (is for); and a extensive (is an extensive)

Line 128: You talked about an acute presentation, what do you consider acute versus chronic? What was the time to onset of clinical signs and diagnosis?  Table 1 specifies the duration of clinical signs, are those from onset to recovery or to diagnosis? In the discussion, you specify less than 16 days for acute presentation. However, you also mentioned that 7/30 dogs had clinical signs for more than 16 days.

Line 146: What is the meaning of B1?

Line 152: Any apparent age predisposition? 

Line 179: Could you describe the residual changes, please?

Line 194:  Were samples taken for histopathological analysis?

Line 298: Any potential connection with a hematogenous spread from other organs (as the primary cause)?

Line 402: Could you discuss the limitations of the study, please? Among the limitations of this study is that all cases come from a referral centre where MRI is available.  Although MRI is considered the modality of choice for diagnosing SEE, in general practice the management and outcome can differ from what has been presented here. Could you mention if there are disadvantages to using MRI?

Line 402:  Any suggestions for future studies?

Author Response

Reviewer 1

Infection can result from either the hematogenous spread of bacteria or by direct local extension, could you determine the source and route of infection in the cases? You mentioned that eight of the 30 dogs had been diagnosed with eight different pathologies between eight weeks and ten days before presentation. Any correlation with the SEE?

The origin of the infection can be suspected in dogs that presented wounds close to SEE or in dogs that were diagnosed with an infectious process. A comment regarding this is included in the Discussion section, lines 297-300.

Any correlation between the follow-up results of the MRI and the improvement of clinical signs? Any prognostic utility?

The aim of performing a follow-up MRI was to have objective additional information to monitor the pathology. According to the reviewer´s comment the text in the Discussion section has been changed to better clarify this point. New lines 394-406: “The imaging showed radiological improvement in all cases, with the absence of previous signs of empyema and osteolysis, and a decrease in paraspinal soft tissue swelling. In the human literature, a correlation between the MRI findings at follow-up and the clinical status has been studied [39], and although no single MRI parameter was associated with the clinical status of patients, it was concluded that soft tissue findings, not bone findings, should be the focus of clinicians interpreting follow-up MRI results. A decrease in soft tissue findings (epidural enhancement, paraspinal inflammation, and disk space enhancement) was considered to be a good indicator of improvement, and the patients whose MRI was equivocal or worse consequently had their antibiotic course prolonged or an invasive intervention more frequently than the patients with improvement on follow-up MR imaging [39]. A follow-up MRI is advised in cases with SEE as it provides valid information on the outcome of these patients. “

Specific comments

Line 76: did conservative management include any debridement of any paraspinal abscess

No, dogs treated conservatively or medically did not receive any kind of surgical procedure. To make this point clear the text has been changed to: “The final decision for treatment was achieved by consensus with the owners after they were informed of the treatment options, namely surgical and medical treatment, or medical treatment alone.” Lines 215-217

Line 85: Were urinary and faecal continence assessed? If so, how many dogs were showing signs of incontinence?

Considering the reviewer´s comment, urinary and faecal continence status have been included in methods section using the modified Frankel Scale

Line 116, 231: Spelling mistake, fot (is for); and a extensive (is an extensive)

Both have been corrected in the text, we apologize for the mistakes

Line 128: You talked about an acute presentation, what do you consider acute versus chronic?

We considered the signs to be acute according to its medical definition of sudden and severe onset. This point has been clarified in the text. Line 127

What was the time to onset of clinical signs and diagnosis?  Table 1 specifies the duration of clinical signs, are those from onset to recovery or to diagnosis?

The time from onset of clinical signs to diagnosis ranged from 2 days to 4 months. Line 148

In the discussion, you specify less than 16 days for acute presentation. However, you also mentioned that 7/30 dogs had clinical signs for more than 16 days.

Although presentation was acute in all dogs, they presented to us at different stages. As in the mentioned human medicine report they discuss the different MRI pattern of less or more than 16 days the same time lapse was considered in order to compare our results.

Line 146: What is the meaning of B1?

Changed to: showed mitral regurgitation B1 (according to the stages of myxomatous mitral valve diseases). Line 154

Line 152: Any apparent age predisposition?

In the 30 dogs in this case series there was no age predisposition, only an average of 6 years. This is mentioned in the discussion in line 294: “…the mean age at presentation of signs was six years,”

Line 179: Could you describe the residual changes, please?

The text has been expanded accordingly, lines 182-183: “…with new bone formation, loss of intervertebral space and mild vertebral marrow edema”

Line 194:  Were samples taken for histopathological analysis?

Text changed to: “As the material obtained was a purulent liquid histopathological studies were not considered, but the samples were submitted for microbial culture.” Lines 205-207

Line 298: Any potential connection with a hematogenous spread from other organs (as the primary cause)?

“For the case of Streptococcus beta-haemolytic, an hematogenous spread from the genitourinary system was suspected, as the dog had undergone orchiectomy for prostatitis 10 days before presentation of signs, for the rest the primary cause remained undetermined.” This has been added to the text. Lines 302-305

Line 402: Could you discuss the limitations of the study, please? Among the limitations of this study is that all cases come from a referral centre where MRI is available.  Although MRI is considered the modality of choice for diagnosing SEE, in general practice the management and outcome can differ from what has been presented here. Could you mention if there are disadvantages to using MRI?

A separate section with the limitations has been included. Lines 426-430.

Regarding the use of MRI we believe this is not a limitation, because as, as the reviewer points out, the MR is the imaging technique of choice for the evaluation of spinal neurological diseases, therefore, in our study an MRI scan showing a space-occupying accumulation of epidural material suggestive of empyema was an inclusion criteria. The fact that all the studies were performed with the same machine and protocols, in our view adds consistency to the results.

Line 402:  Any suggestions for future studies?

The text has been expanded, lines 429-432: Further studies are needed with larger and more homogeneous numbers of surgically and conservatively treated dogs, and with full complementary tests including haematological cultures, in order to identify the most common pathogens and to elaborate treatment guidelines.

Reviewer 2 Report

Dear authors,

Thank you for your submission. It is of interest but needs attention.

You title seems to be incorrect. The surgery was always combined with medical treatment. So it is not 'or'

No remarks concerning the abstract, introduction.

Line 63: Concerning the 'method' section I suggest a revision. First of all it is a retrospective study and you are suggesting that it is prospective. f.i.: were all dogs anesthetized the same way? This section could be removed anyway as it is not informative for this study. Just say that they were induced and were under general anesthesia.

And in a retrospective study line 108 does not sound logical.

line 83: The reference of Scott is inappropriate. The original reference is from prof Griffiths (Glasgow University). It dates from many years ago. It is not from Scott.

line 120 and afterwards line 259. This does not sound valid. You can easily perform, even on these number, f.i. a wilcoxon test. But in your discussion you need to answer the validity of your result.  

line 136: antimicrobials. Which ones as you are continuing with them.

line 143: in one dog CRP was tested,... So it is retrospective as this was not routinely done earlier.

Table 1 and 2 are not informative. I suggest simple text for table 1. And table 2 suggests that you are discussion the MRI features, but you are not doing that. There is a brief section in the text. And with these low number of patients, I suggest to remove table 2. Expand the section in the text.

Figures 2 and 3: is it possible to colour the arrows?

Line 265: discussion

Your discussion can be shortened. It is now too lengthy, and some parts can be deleted as they are not informative. F.i. line 298 to 314 can be denser. And a discussion is a critical analysis of your observations. I miss a critical analysis and 'limitations of this study'.

Line 288 to 290: what are you saying here. It doesn't make sense.

Line 315 to line 322 could be shortened to f.i.: although in this retrospective study only in one dog CRP was measured, based on recent studies, it should be part of the diagnostic protocol in cases suggestive of SEE (5).

Line 334 write SEE

Line 352: use another word for controversial. Whether you choose a medical or surgical treatment is not controversial.

Line 366: you are trying to make an algorithm here. A sound and logical reasoning. Financial constraints do not belong here. We suggest a proper treatment regardless of the situation. However, financial constraints can be, afterwards, a reason to choose differently.

Line 367-374: please rewrite. We treat based on our knowledge of a situation and our observations. The part of the antibiotic resistance does not belong here. If you want to discuss it: limitations of the study and recommendations.

Line 375-382 please rewrite.

line 381: spinal canal occupation does not exist. Occupation is a job. Please use spinal cord compression.

Line 388: this is n=2 and does not justify this conclusion.

Line 389: occupation

line 401-402: You can not write this. I suggest: a follow-up MRI is advised in cases with SEE as it provides valid information on the outcome of these patients.

Line 403: Conclusion

Line 404: remove the comma. New sentence: Sadly, a delay in diagnosis etc

Line 407: you can not write that surgical treatment least to a more rapid improvement. First of all, you numbers are too low to justify such a statement. If you want to prove this you need two comparable groups of patients: one treated medically and the other one treated medically with surgery.

Line 411-414: remove to limitations of this study (and add this section).

Author Response

Reviewer 2

You title seems to be incorrect. The surgery was always combined with medical treatment. So it is not 'or'

The title has been changed according to the reviewer´s comment to: “

Clinical presentation, MRI characteristics, and outcome of conservative or surgical management of spinal epidural empyema in 30 dogs

Line 63: Concerning the 'method' section I suggest a revision. First of all it is a retrospective study and you are suggesting that it is prospective. f.i.: were all dogs anesthetized the same way? This section could be removed anyway as it is not informative for this study. Just say that they were induced and were under general anesthesia.

Changes have made according to the reviewer´s comment.

Lines 70-71: “…follow-up including MRI results if performed,…”

The text regarding treatment options has been changed and moved to results. Lines 204-208

Lines 76-77: “The neurological status at the time of presentation and during treatment, at the time of discharge, and at re-examinations every three weeks was recorded.”

Line 84: “Follow-up had to be for a minimum of 3 months”

Lines 90-92: “MRI imaging studies of the vertebral column were acquired with a 1.5 Tesla unit (Gyroscan Intera, Philips, Eindhoven, The Netherlands). All studies were performed with the dogs in dorsal recumbency under inhalation anaesthesia.“

And in a retrospective study line 108 does not sound logical.

See previous answer. Line 71: “…follow-up including MRI results if performed”

line 83: The reference of Scott is inappropriate. The original reference is from prof Griffiths (Glasgow University). It dates from many years ago. It is not from Scott.

Regarding the comments of both reviewers and the characteristics of the cases, we believe that the most appropriate scale would be the modified Frankel scale, therefore we have changed the text accordingly.

line 120 and afterwards line 259. This does not sound valid. You can easily perform, even on these number, f.i. a wilcoxon test. But in your discussion you need to answer the validity of your result. 

According to the comment the text in the Statistical section has been changed to better clarify this point. Lines 116-119, 278-283 and in the discussion section line 292-294.

line 136: antimicrobials. Which ones as you are continuing with them.

The information is now included in the text. Lines 141-145

line 143: in one dog CRP was tested,... So it is retrospective as this was not routinely done earlier.

Indeed, a comment on this is included in the discussion.

Table 1 and 2 are not informative. I suggest simple text for table 1. And table 2 suggests that you are discussion the MRI features, but you are not doing that. There is a brief section in the text. And with these low number of patients, I suggest to remove table 2. Expand the section in the text.

Descriptive paragraph in place of table 1:

“The median age was 6,1 years (range of 8 months-11 years). Sixteen dogs were male (9 neutered) and 14 were female (10 neutered). The average weight was 24,4 kg (range 3,6-50 kg). Breeds comprised American Staffordshire Terrier (n=5), Labrador Retriever (n=3), Doberman (n=2), Dalmatian (n=1), Yorkshire Terrier (n=2), German Shepard (n=2), Golden Retriever (n=2), French Bulldog (n=1), West Highland white terrier (n=1), English Bulldog (n=1), Maltese (n=1), crossbreed (n=5), Greyhound (n=2), English Pointer (n=1) and Dachshund (n=1).”

Descriptive paragraphs in place of table 2:

“At presentation, general physical examination revealed abnormalities in 28/30 dogs, including pain (28/30), pyrexia (3/30), chronic dermatitis (1/30) and skin wounds (1/30). Neurological examination showed abnormalities in 22/30 dogs, which included ambulatory paraparesis (13/22), non-ambulatory paraparesis (3/22), paraplegia with preserved deep pain sensation (3/22), ambulatory monoparesis (1/22), ambulatory tetraparesis (1/22) and nerve root sign (1/30).”

“The images revealed the presence of a space-occupying accumulation of epidural material suggestive of empyema localized in the spinal segments C1-C5 (6/30), C6-T2 (2/30), T3-L3 (11/30), or L4-S3 (11/30). In 7/30 dogs the epidural material was extensive, involving from 2 to up to 9 consecutive vertebral bodies. In addition, 15/30 dogs had signs of discospondylitis at adjacent levels, and in 13/30 dogs there was involvement of adjacent soft tissues and vertebral bone. Only in 2/30 dogs empyema was the only finding. “

“In the conservatively treated group, the most severe cases included 1 dog with non-ambulatory paraparesis and 2 dogs that had paraplegia with preserved deep pain sensation for 24-48 hours. Regarding the imaging, 7/24 had extensive epidural occupation with involvement of 2 intervertebral levels up to 9 consecutive vertebral bodies, or had spinal cord compression of 25-50%.”

Figures 2 and 3: is it possible to colour the arrows?

we attach figures with arrows in different colours

Line 265: discussion

Your discussion can be shortened. It is now too lengthy, and some parts can be deleted as they are not informative. F.i. line 298 to 314 can be denser. And a discussion is a critical analysis of your observations. I miss a critical analysis and 'limitations of this study'

The text has been changed, in new lines 322-330) to: “All six samples collected during surgery showed bacterial growth with 2/6 cases positive for Pseudomona aeruginosa and each of the remaining cases positive for Aerococcus spp, Serratia marcescens, Burkholderia cepacia, Streptococcus beta-hemolyticus. For the case with Streptococcus beta-haemolytic infection, an hematogenous spread from the genitourinary system was suspected, as the dog had undergone orchiectomy for prostatitis 10 days before the presentation of signs, for the rest the primary cause remained undetermined. The only pathogen in common with previous studies is Pseudomona aeruginosa [12]. In human patients, gram-positive Staphylococcus aureus is the most common agent [5, 1, 26]. Our case series shows increased identification of causative agents in surgical specimens as described in a recent SEE study [12] and human medicine [26,27]. The results obtained in the present case series do not allow a conclusion on the most common causative agent due to the low number of positive cultures and the variability of the results.”

Another changes have been done throughout the discussion section.

A new section with the limitations has been included.

Line 288 to 290: what are you saying here. It doesn't make sense

The text has been changed, new lines 283-288: In human medicine, most of the patients present as well back pain (71%) and pyrexia (66%) as the initial symptoms [1]. Apart from these signs, a progressive neurologic dysfunction has been reported in both humans and dogs with SEE [3, 4, 5, 12]. In human medicine, this triad is only present in 10–30% on initial presentation and correlates to advanced stages, which may lead to a significant diagnostic delay in about 75% of patients [5]. Similarly, in our group of dogs, only 10% of them presented the triad, therefore, in an attempt to avoid diagnostic delays, this pathology must be considered despite not having the reported triad of signs.

Line 315 to line 322 could be shortened to f.i.: although in this retrospective study only in one dog CRP was measured, based on recent studies, it should be part of the diagnostic protocol in cases suggestive of SEE (5)

The text has been shortened and changed accordingly. Lines: 312-314

“Although CRP was only measured in one dog in this retrospective study, based on recent studies [28], it should be part of the diagnostic protocol in cases suggestive of SEE.”

Line 334 write SEE

Changed, new line: 324

Line 352: use another word for controversial. Whether you choose a medical or surgical treatment is not controversial.

The text has been changed to: “Regarding treatment, the choice of medical or surgical treatment of SSE remains debatable in veterinary and human medicine.” Line 364

Line 366: you are trying to make an algorithm here. A sound and logical reasoning. Financial constraints do not belong here. We suggest a proper treatment regardless of the situation. However, financial constraints can be, afterwards, a reason to choose differently

The reference to financial constrains has been deleted.

Line 367-374: please rewrite. We treat based on our knowledge of a situation and our observations. The part of the antibiotic resistance does not belong here. If you want to discuss it: limitations of the study and recommendations.

The text has been changed to. “The medical treatment used in this case series for dogs in which the causative agent was not identified, was a combination of antimicrobials based in our knowledge of a situation and our observations.”. Lines 376-378

A comment on antibiotic resistance is now included in limitations.

Line 375-382 please rewrite

Text changed to: “Surgical treatment in this case series was indicated for dogs with ≥25-50% spinal cord compression. In human medicine, surgical treatment is established based on the degree of compression of >50% on cross-sectional MRI images [37], which correlates with severe motor neurological deficits. This approach shows favorable results in patients with deficits of less than 36 hours duration [38].” Lines 379-83

line 381: spinal canal occupation does not exist. Occupation is a job. Please use spinal cord compression.

Changed

Line 388: this is n=2 and does not justify this conclusion

Changed to: “However, the risks of general anaesthesia and surgery must be considered, as well as the possible favourable outcome with non-surgical treatment.”. Line: 387-388

Line 389: occupation

Deleted

Line 401-402: You can not write this. I suggest: a follow-up MRI is advised in cases with SEE as it provides valid information on the outcome of these patients

Changed to: “A follow-up MRI is advised in cases with SEE as it provides valid information on the outcome of these patients.” Lines 405-406

Line 403: Conclusion

Changed

Line 404: remove the comma. New sentence: Sadly, a delay in diagnosis et

Changed to: “SEE is a neurological emergency that requires early diagnosis and appropriate treatment as it is potentially fatal. Sadly, a delay in diagnosis is not uncommon due to the unspecific clinical signs.” Line 410-411

Line 407: you can not write that surgical treatment least to a more rapid improvement. First of all, you numbers are too low to justify such a statement. If you want to prove this you need two comparable groups of patients: one treated medically and the other one treated medically with surgery.

Changed to: “Surgical treatment, considering the limited number of cases, tends to…”

Line 411-414: remove to limitations of this study (and add this section)

A new section has been included

“Limitations:

The retrospective nature of the design and the limited number of surgical cases may cause a bias in this study. Another limitation of this study would be the lack of haematological cultures, which may help in pathogen identification and avoid antimicrobials combinations that may cause future bacterial resistances.

Round 2

Reviewer 2 Report

Thank you for your revision. It has much improved now. It is a valid study. My only remarks concerns the limitations section. I suggest to change it into:

This is a retrospective study analysing cases seen within a large time frame. Ideally CRP had been measured in all cases, and haematological cultures obtained to identify pathogens and avoid antimicrobials combinations that may cause bacterial resistance. And the number of surgical cases is, compared to the conservatively treated cases, very small. However we believe that this analysis of seen cases helps in understanding and treating these cases.

Author Response

Change have made according to the reviewer´s comment

 This is a retrospective study analysing cases seen within a large time frame. Ideally CRP had been measured in all cases, and haematological cultures obtained to identify pathogens and avoid antimicrobials combinations that may cause bacterial resistance. And the number of surgical cases is, compared to the conservatively treated cases, very small. However, we believe that this analysis of seen cases helps in understanding and treating these cases
